# Structural Modifications of Nature-Inspired Indoloquinolines: A Mini Review of Their Potential Antiproliferative Activity

**DOI:** 10.3390/molecules24112121

**Published:** 2019-06-05

**Authors:** Ning Wang, Marta Świtalska, Li Wang, Elkhabiry Shaban, Md Imran Hossain, Ibrahim El Tantawy El Sayed, Joanna Wietrzyk, Tsutomu Inokuchi

**Affiliations:** 1Division of Chemistry and Biotechnology, Graduate School of Natural Science and Technology, Okayama University, 3-1-1 Tsushima-naka, Kita-ku, Okayama 700-8530, Japan; wn12171982@sina.com (N.W.); liwang_512@163.com (L.W.); shaban_nrc@yahoo.com (E.S.); imranju7@yahoo.com (M.I.H.); 2Department of Materials Science and Engineering, Yingkou Institute of Technology, Yingkou 115014, China; 3Hirszfeld Institute of Immunology and Experimental Therapy, Polish Academy of Sciences, 12R. Weigla Street, 53-114 Wroclaw, Poland; switalska@iitd.pan.wroc.pl; 4Department of Medicinal Chemistry, School of Pharmacy, Southwest Medical University, Luzhou 646000, China; 5Dyeing, Printing and Textile Auxiliaries Department, Textile Research Division, National Research Centre, 33 El-Bohouth Street, Dokki, Giza 12622, Egypt; 6Department of BioMolecular Sciences, School of Pharmacy, University of Mississippi, 419 Faser Hall, MS 38677, USA; 7Chemistry Department, Faculty of Science, El Menoufia University, Shebin El Koom 32511, Egypt

**Keywords:** cryptolepine, neocryptolepine, isocryptolepine, antiproliferative activity, structure activity relationships

## Abstract

Cryptolepine, neocryptolepine and isocryptolepine are naturally occurring indoloquinoline alkaloids with various spectrum of biological properties. Structural modification is an extremely effective means to improve their bioactivities. This review enumerates several neocryptolepine and isocryptolepine analogues with potent antiproliferative activity against MV4-11 (leukemia), A549 (lung cancer), HCT116 (colon cancer) cell lines in vitro. Its activity towards normal mouse fibroblasts BALB/3T3 was also evaluated. Furthermore, structure activity relationships (SAR) are briefly discussed. The anticancer screening of neocryptolepine derivatives was performed in order to determine their cytotoxic and growth inhibitory activities across the JFCR39 cancer cell line panel.

## 1. Introduction

### 1.1. Antitumoral Activity of Cryptolepine

Cryptolepine (**1**), neocryptolepine (**2**), and isocryptolepine (**3**) are typical indoloquinoline alkaloids isolated from the roots of *Cryptolepis sanguinolenta* [1,2].

These alkaloids are composed of the tetracyclic indoloquinoline ring system, which only differ with respect to the orientation and site of their indole and quinoline ring junctures (Figure 1). By introducing the appropriate motif at the certain positions, their derivatives can increase better biological activity than the mother core [3]. 

Many researchers have achieved the structural modification of cryptolepine scaffold for the purpose of improving antitumor activity, while the pharmacological properties of the analogues are being studied deeply. 

The bioactivities of many antitumoral agents are related to their interactions with the DNA molecule, which is regarded as a classical target for these drugs in clinical use. The basic mechanism of antitumoral activity of these drugs is to affect the replication, expression, transcription and other physiological functions of the DNA, which causes the tumor cell death [4].

In 1990, Yamato et al. synthesized the indoloquinoline derivatives **4** (Figure 2), and screened its biological properties in vitro and in vivo. The compounds **4** showed potential antitumor activity (P388 leukemia in mice), DNA intercalative property, and ability to induce topoisomerase II dependent DNA cleavage [5].

In 1997, Deady et al. studied a series of cryptolepine derivatives **5** and evaluated their antitumoral activity in a series of murine and human tumor cell lines such as the mice lung cancer cells (LLC), mice leukemia cells (P388), human leukocyte cells (JL). These compounds appear to be mixed topoisomerase I/II inhibitors in the human leukemia cell lines studied [6].

In 1998, Bonjean et al. verified the cryptolepine alkaloids bound tightly to DNA as a typical intercalating agent by various means of absorption, such as fluorescence, circular, and linear dichroism, as well as by a relaxation assay using DNA topoisomerases. At the same time, they provided direct evidence that DNA is the primary target of cryptolepine. The mechanism of the compounds inhibiting tumor cell proliferation is mainly based on the synthesis of DNA inhibition, not the inhibition of proteins and RNA [7]. 

In 2002, Lisgarten, John N., reported that cryptolepine interacts with the DNA fragment d(CCTAGG)_2_ in a base-stacking intercalation mode by using X-ray crystallography. It was found that cryptolepine intercalated between pyrimidine bases of the fragment in the form of π-π accumulation. This is the first single crystal structure of DNA intercalator complex, which is the small molecule to bind a non-alternating (pyrimidine-pyrimidine) DNA sequence [8].

In 2012, Boddupally et al. have synthesized a series of 11-substituted cryptolepine derivatives **6** (Figure 2). The compound **6** showed the most potent anticancer activity with IC_50_ = 0.97 μM against HCT-116 colon cancer cell line and IC_50_ = 2.33 μM against Raji lymphoma cells in further cytotoxic test in vitro. At the same time, this compound showed a strong inhibition of c-MYC expression [9].

Gu and Lu demonstrated the binding of aniline-substituted cryptolepine derivatives with calf-thymus DNA presumably via an intercalation mechanism and studied the binding mode of these derivatives with duplex DNA by Surflex-dock software. They reported that these derivatives intercalated into the base-pairs, and reacted with DNA via mainly π-π interaction with medium, moreover the functional groups substituted on aniline ring affected the binding abilities [10,11]. 

The aim of this review is to present an overview of the potential of neocryptolepine and isocryptolepine as scaffolds for the design and development of new anticancer drugs. Both compounds have also the linearly arranged tetracyclic plane as same as cryptolepine, so they can be expected as candidates of antitumoral agent, although they have a slightly weaker capability to intercalate into DNA and inhibit human topoisomerase II [12].

### 1.2. Antitumoral Activity of Neocryptolepines and Isocryptolepines 

In our group, we have synthesized an array of novel neocryptolepine derivatives **7**, and their congeners **8**, 11-aminoalkylamino-substituted chromeno[2,3-*b*]indoles **9**, and isocryptolepine derivatives **10** (Figure 3). Then the antiproliferative activities of these compounds were tested in vitro against MV4-11 (human leukemia), HCT116 (human colon cancer), and A549 (human non-small cell lung cancer) and BALB/3T3 (normal murine fibroblasts) cell lines. 

We focused our attention on modifying the neocryptolepine structure at four positions: (1) the side chain at C11, introducing the various amino groups, e.g., -NH(CH_2_)_n_NH_2_ (n = 2,3,4), -NH(CH_2_)_3_N(C_2_H_5_)_2_, -NH(CH_2_)_3_OH, -NHCH_2_CH(OH)CH_2_NH_2_, -NHCH_2_CH(CH_3_)CH_2_NH_2_, -NHCH_2_CH(CH_3_)NH_2_, -NHCH_2_C(CH_3_)_2_NH_2_, and so on; (2) replacing the H atom at C2 with MeO, Br, Cl, Me, and replacing H at C9 with COOMe; (3) altering the Me from N5 to N6, (4) changing the N5 to O atom and (5) incorporating metal ion coordinated with diamine side chain [13,14,15,16,17,18].

In addition, we have also engaged in modifying isocryptolepine derivatives in three ways: the amino substituent effect at C6, and N11 methyl localization effect, and the substituent group effect at C2 of quinoline moiety [19]. 

Some compounds with higher antiproliferative activity are listed in Table 1 and Table 2. The SAR (structure activity relationships) studies reveal that the most necessary strategy is introducing ω-aminoalkylamino group in the side chain. Among the tested 84-neocryptolepine derivatives and 44-isocryptolepine derivatives, the most potent compounds with higher activity contain the -NH(CH_2_)_3_NH_2_ side chain. 

Among them, 11-(3-amino-2-hydroxy)propylamino derivatives **7e** and **7f** were the most cytotoxic with a mean IC_50_ value of 0.042 μM and 0.057 μM against the MV4-11 cell line, 0.197 μM and 0.190 μM against the A549 cell line, and 0.138 μM and 0.117 μM against the HCT116 cell line.

We propose the reason that the amino terminus domain in the side chain would react with negative charged phosphate groups in DNA, which increases the insertion ability of the complex to DNA.

The methyl localization effect is an important additional contributor to increase the activity. The activity is greatly reduced (10~30 fold lower) by changing the Me group from N5 to N6 (i.e., **8**) of neocryptolepine derivatives. The Me group can increase the activity of isocryptolepine derivatives, as found by a comparison of the compounds **10e** and **10c**. 

Replacing H atom at the C2 with MeO, Br, Cl, Me, or replacing H at C9 by COOMe can improve the antiproliferactive activity of neocryptolepine derivatives. Each group contributes differently to the antiproliferactive activity of different tumor cells. For example, **7c** with Br at C2 shows the best antiproliferactive activity against MV4-11 leukemia cells, and this modification did not increase antiproliferactive activity against normal BALB/3T3 cells, but unexpectedly it decreases 2-fold the antiproliferactive activity against A549 lung cancer cells. It can be suggested that this modification increases the selectivity of the tested compound, improving their effect on leukemic cells, without increasing possible adverse effects on normal cells. The substituent effect at C2 of isocryptolepine core also can change the activity; NO_2_ is the effective group to increase the activity against cancer but unfortunately also normal cells (**10c** and **10e**). Therefore, we must examine our best to find an effective substituent for further improving the activity.

We boldly tried to replace the 5-nirtogen atom with oxygen atom, forming **9**, which significantly drop the antiproliferactive activity [16].

From all the assay data, isocryptolepine analogues are more potential as the anticancer drug candidates in comparison with the neocrytolepine analogues, compound **10e** (or **10a**) shows the higher antiproliferactive activity against A549 and HCT116 cancer cell, and lower antiproliferactive activity against normal cells comparing **7a**, when amino-substituent is NH(CH_2_)_3_NH_2_ and substituent group is H at C2.

The mode of neocryptolepine and isocryptolepine binding to DNA was studied using UV-VIS absorption spectroscopy with salmon fish sperm DNA. From the DNA binding studies, it can be proven that the methyl localization effect and the substituent group effect at C2 of quinoline moiety influence the capability to intercalate into DNA. Two effects improve the activity of isocryptolepine to interact with DNA, the binding constant of **10f**-DNA was 1.05 × 10^6^ L/mol and **10g**-DNA was 4.84 × 10^6^ L/mol [19]. The activity of neocryptolepine to interact with DNA varied; the binding constant of **7c**-DNA was 2.93 × 10^5^ and **8**-DNA was 3.28 × 10^5^ L/mol [13].

In a series of neocrytolepine analogues, **7c** shows the best antiproliferactive activity against MV4-11 leukemia with IC_50_ 0.012 ± 0.002 μM. Compounds **7f** and **7i** show almost the same highest antiproliferactive activity against A549 lung cancer cells with IC_50_ 0.190 ± 0.027 μM and 0.194 ± 0.063 μM, and the antiproliferative activity against HCT116 colon cancer cells with IC_50_ 0.117 ± 0.055 μM and 0.116 ± 0.078 μM. Analogue **7b** unfortunately shows 2-fold higher cytotoxic activity against normal BALB/3T3 cells with the IC_50_ 0.401 ± 0.015 μM than other studied neocrytolepine analogues and 20-fold higher than cisplatin [13]. 

In series of isocryptolepine analogues, **10e** shows the best antiproliferactive activity against cancer cell lines with IC_50_ 0.05 ± 0.01 μM (MV4-11), IC_50_ 0.11 ± 0.07 μM (A549), and IC_50_ 0.01 ± 0.00 μM (HCT116). Unfortunately, this compound had also 4~15 fold higher cytotoxic activity against normal BALB/3T3 cells with the IC_50_ 0.08 ± 0.02 μM. These two simultaneous modifications at C2-NO_2_ and at N11-Me showed 15- and 100-fold increased cytotoxicity of **10e** in comparison to doxorubicin and cisplatin, cytostatic used as a control.

In 2016, in vitro antiproliferative activities of neocryptolepine derivatives were evaluated by Okada (Table 3). The compounds **7a** and **7d** showed the highest anticancer activity, having IC_50_ values of 0.50 μM and 0.20 μM against the breast cancer MDA-MB-453 cell line, at the same time **7a** having 0.64 μM and 2.7 μM against WiDr (colon adenocarcinoma) and SKOv3 (ovarian cancer) cell lines, **7d** having 0.37 μM and 1.3 μM against WiDr (colon adenocarcinoma) and SKOv3 (ovarian cancer) cell lines.

Recently, we reported the in vitro antiproliferative activity of 11-substituted neocryptolepines with branched ω-aminoalkylamino chain. These 2-substituted 5-methyl-indolo[2,3-*b*]quinoline derivatives were prepared by nucleophilic aromatic substitution (S_N_Ar) of 11-chloroneocryptolepines **2** with appropriate 1,2- and 1,3-diamines (Scheme 1). Many of the prepared neocryptolepine derivatives showed submicromolar antiproliferative activity of less than μM against the human leukemia MV4-11 cell line (Table 4). 

For further studying cytotoxic and growth inhibitory activities of neocryptolepine derivatives, the anticancer screening was performed across the JFCR39 cancer cell line panel, evaluated at five concentration levels (100, 10, 1.0, 0.1, and 0.01 µM) (Table 5). The results (the means of GI_50_, TGI, and LC_50_ values) showed that **7d** have potent anticancer activity against the melanoma (LOX-IMVI) and lung (NCIH522, A549, and DMS273) cancer cell lines in the JFCR39 panels. Compound **7d** shows a notable cytotoxicity against the breast (BSY-1, LC_50_ = 0.82 µM), CNS (SF-539, LC_50_ = 0.60 µM, SNB-75, LC_50_ = 0.91 µM), colon (HCC2998, LC_50_ = 0.74 µM), lung (NCI-H522, LC_50_ = 0.71 µM, A549, LC_50_ = 0.85, DMS273, LC_50_ = 0.70, DMS114, LC_50_ = 0.72 µM), and ovarian (VCAR-4, LC_50_ = 0.48 µM) cancer cell lines [20].

Table 5 shows that all of the compounds are highly potent against the melanoma (LOX-IMVI) and lung (NCI-H522, A549, and DMS273) cancer cell lines in the JFCR39 panels. Compounds **7d** and **7k** are more potent than compound **7q** towards the cancer cell lines in the JFCR39 panel. Compounds **7d** and **7k** show a similar growth inhibitory activity (GI_50_ > 0.01 µM) against the 31 and 28 cell lines, respectively, of the JFCR39 panel. Compound **7k** shows a notable cytotoxicity against the breast (BSY-1, LC_50_ = 0.82 µM), CNS (SF-539, LC_50_ = 0.60 µM, SNB-75, LC_50_ = 0.91 µM), colon (HCC2998, LC_50_ = 0.74 µM), lung (NCI-H522, LC_50_ = 0.71 µM, A549, LC_50_ = 0.85, DMS273, LC_50_ = 0.70, DMS114, LC_50_ = 0.72) and ovarian (VCAR-4, LC_50_ = 0.48 µM) cancer cell lines. Compound **7q** shows a mentionable cytotoxicity for the CNS (SF-539, LC_50_ = 0.63 µM, SNB-75, LC_50_ = 0.93 µM), colon (HCC2998, LC_50_ = 0.69 µM), lung (NCI-H522, IC_50_ = 0.57 µM, DMS273, LC_50_ = 0.61 µM), and melanoma (LOX-IMVI, LC_50_ = 0.75 µM) cancer cell lines. Compound **7q** displayed a strong growth inhibitory activity (GI_50_ < 1 µM) for 28 panels in the JFCR39 panel of the various cell lines. DMS273 (lung cancer) and NCI-H522 (lung cancer) are the most sensitive cell lines of the 39 cell lines (GI_50_ = 0.42 µM for both and LC_50_ = 4.7 µM and 5.5 µM respectively). Compound **7q** also showed a strong antitumor effect on LOX-IMVI (melanoma, GI_50_ = 0.3 µM, LC_50_ = 7.6 µM).

In our group, we achieved the synthesis of the artemisinin-indoloquinoline hybrids and studied their antiproliferative activity (Figure 4). The artemisinin-neocryptolepine hybrids **11a**, **11b** and the artemisinin-isocryptolepine hybrids **12** showed the potent antiproliferative activity, based on the data in Table 6.

In 2017, Li Wang synthesized the novel tacrine-neocryptolepine hybrids, and evaluated as inhibitors of human AChE (hAChE) and human BuChE (hBuChE) (Figure 5). Compound **13** was a highly potent hAChEI having IC_50_ = 0.95 ± 0.04 nM, and a highly potent hBuChE having IC_50_ = 2.29 ± 0.14 nM [22].

## 2. Compare Study

The COMPARE analysis assesses the correlation coefficient between the fingerprints of the test compounds and those of the various reference compounds [23]. This system provides an information intensive approach to identify the molecular targets of new compounds. The JFCR39 COMPARE analysis-guided assay is a successful means to find new anticancer drug candidates. The COMPARE analysis is carried out by calculation of the Pearson correlation coefficient (r value) between the fingerprints of compounds X and Y. The r value is then used to determine the degree of similarity, that is, the higher the r value, the greater the similarity of X to Y. Generally, an r value of 0.5 < r < 0.75 between two agents suggests they might have a similar mechanism of action (Figure 6).

The COMPARE analysis revealed that compounds **7d** and **7k** have a very good match to actinomycin D (r = 0.7 for both). Similarly, compound **7k** has a slight similarity to paclitaxel (r = 0.64). Compound **7q** shows some resemblance to vindesine sulfate (r = 0.58) and aclarubicinHCl (r = 0.57). 

## 3. Conclusions

Indoloquinoline alkaloids are important scaffolds for antitumoral drug development. This review discussed the SAR of neocryptolepine and isocryptolepine, and presents the useful methods for improving the antitumoral activity of neocryptolepine and isocryptolepine analogues. The amino substituent effect and methyl localization effect are now available strategies, but the substituent group effect at the benzene ring of the quinoline moiety has no effect on regular SAR. Thus, the antitumoral activity is highly related to the activity of interacting with DNA. The computer-assisted database analysis, COMPARE, suggested that **7d** and **7k** have a mode of action similar to actinomycin D. It also suggested that **7l** has a mode of action similar to vindesine sulfate or aclarubicin HCl. However, the new compounds may have other unique modes of action since the correlation coefficients (r) were at relatively low levels, which present an interesting possibility to examine in further studies.

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
