# Peer review of "Structural Modifications of Nature-Inspired Indoloquinolines: A Mini Review of Their Potential Antiproliferative Activity"

_molecules, 2019, doi:10.3390/molecules24112121_

Round 1
Reviewer 1 Report
The review by Wang and colleagues briefly describes the history of modification of the cryptolepine scaffold to produce anticancer drugs and tabulates cytotoxicity data. Many of the compounds are highly cytotoxic toward a variety of cultured cells. At this point, it seems prudent to move beyond cell culture to determine if these compounds are efficacious in vivo rather than to continue to tune for incremental changes in IC50 values.
The data in Figure 2 are of low resolution and of unclear utility.
Author Response
Answer to comments (Molecules).
Firstly, we would like to express sincere thanks to Reviewers for careful reading of our manuscript and giving us courteous comments to improve the Manuscript. We have revised the manuscript in order to meet with reviewers’ opinion and suggestions. Enclosed are the revised manuscript and explanation for the revision. We would like to submit the revised manuscript here. I appreciate your favorable considerations and judgments.
[1] Author's Reply to the Review Report (Reviewer 1)
Comments and Suggestions for Authors: The review by Wang and colleagues briefly describes the history of modification of the cryptolepine scaffold to produce anticancer drugs and tabulates cytotoxicity data. Many of the compounds are highly cytotoxic toward a variety of cultured cells. At this point, it seems prudent to move beyond cell culture to determine if these compounds are efficacious in vivo rather than to continue to tune for incremental changes in IC50 values.
The data in Figure 2 are of low resolution and of unclear utility.
[Reply to reviewer 1]
Thank you for your kind suggestions for our continuous study with Indoloquinolines.
The compounds in Figure 2 are examples reported by other authors and typical examples were introduced from the literatures.
Reviewer 2 Report
The manuscript gives an excellent review on a SAR study on antiproliferative indoloisoquinoline-based alkaloid analogues that merits publication in Molecules, however prior to final acceptance the following points must be addressed: (i) the pi-pi interaction associated with DNA intercalation should be discussed in more detail with special regard to identification of donor and acceptor sites; (ii) a picturial presentation, if available, would be beneficial to illustrate the nature of the more complex interaction involved in the crucial intercalation; (iii) the influence of the length of the basic side chain to the antiproliferative should also deserve a bit more attention; (iv.) spelling needs revision throughout the manuscript, e.g. the title should be corrected.
Author Response
Firstly, we would like to express sincere thanks to Reviewers for careful reading of our manuscript and giving us courteous comments to improve the Manuscript. We have revised the manuscript in order to meet with reviewers’ opinion and suggestions. Enclosed are the revised manuscript and explanation for the revision. We would like to submit the revised manuscript here. I appreciate your favorable considerations and judgments.
[2] Author's Reply to the Review Report (Reviewer 2)
Comments and Suggestions for Authors: The manuscript gives an excellent review on a SAR study on antiproliferative indoloisoquinoline-based alkaloid analogues that merits publication in Molecules, however prior to final acceptance the following points must be addressed: (i) the pi-pi interaction associated with DNA intercalation should be discussed in more detail with special regard to identification of donor and acceptor sites; (ii) a picturial presentation, if available, would be beneficial to illustrate the nature of the more complex interaction involved in the crucial intercalation; (iii) the influence of the length of the basic side chain to the antiproliferative should also deserve a bit more attention; (iv.) spelling needs revision throughout the manuscript, e.g. the title should be corrected.
[Reply to reviewer 2]
Tittle was modified:
Structural Modifications of Nature-Inspired Indoloquinolines: A Mini Review of Their Potential Antiproliferative Activity
(i) The pi-pi interaction associated with DNA intercalation should be discussed in more detail with special regard to identification of donor and acceptor sites:
Discussion of interaction associated with DNA intercalation is described in page 3 and our results are described in page 5:
“The mode of neocryptolepine and isocryptolepine binding to DNA was studied using UV-Vis absorption spectroscopy with salmon fish sperm DNA. From the DNA binding studies, it can be proven that the methyl localization effect and the substituent group effect at C2 of quinoline moiety influence the capability to intercalate into DNA. Two effects improve the activity of isocryptolepine to interact with DNA, the binding constant of 10f-DNA was 1.05×106 L/mol and 10g-DNA was 4.84×106 L/mol [17]. And the activity of neocryptolepine to interact with DNA varied; the binding constant of 7c-DNA was 2.93×105 and 8-DNA was 3.28×105 L/mol [13].”
In future we will try the pi-stacking of indolo[2,3-b]quinoline core and electrostatic interactions of the amine substituents at C11.
(ii) a picturial presentation, if available, would be beneficial to illustrate the nature of the more complex interaction involved in the crucial intercalation;
We have no picturial presentations about DNA interaction, but we have reported data of (a) UV-Vis absorption spectra of compound 7c and 7c-DNA at 20 oC. in reference [13].
(iii) the influence of the length of the basic side chain to the antiproliferative should also deserve a bit more attention;
Examples by changing the length of omega-amnioalkylamino substituents are listed in Table 1, but only two example; NH(CH2)2NH2., NH(CH2)3NH2.
(iv.) spelling needs revision throughout the manuscript, e.g. the title should be corrected.
Mistake1: 114 A549 cell line, and 0.138 μM and 0.117 μM against the BALB/3T3 cell line.
Coreection: “BALB/3T3” was changed to “HCT116 ”
Mistake2: 147 activity against HCT 116 colon
Coreection: “ HCT 116” was changed to “HCT116 ” , remove the “space character “
Mistake3: 151 lines with IC50 0.05±0.01 μM (MV4-11), IC50 0.11±0.07 μM (A549), and IC50 0.01±0.00 μM (HCT 116).
Coreection: “ HCT 116” was changed to “HCT116 ” , remove the “space character “
Mistake3: From line 155 to 159 double space between lines
Coreection: Line Spacing chage to 13 pounds
Mistake4: 156 (Table 3). The compounds 7a and 7d showed the highest anticancer activity in Table 3,
Coreection: remove “in Table 3”
Mistake5,6:
158 WiDr (colon adenocarcinoma) and SKOv3 (ovarian cancer) cell line, 7d having 0.37 μM and 1.3 μM against
159 WiDr (colon adenocarcinoma) and SKOv3 (ovarian cancer) cell line.
Coreection: “cell line”was changed to “cell lines ”
Mistake7:
174 Table 4. Antiproliferative activity of 11-aminoalkylamino-5-methyl- indolo[2,3-b]quinolines against cell line of
175 MV4-11, A549 and HCT116, and cytotoxicity against normal mice fibroblast BALB/3T3
Coreection:“cell line”was changed to “cell lines ”
Mistake 8: in Table 5 “MDA-MB-23”
Coreection: “MDA-MB-23” was changed to “ MDA-MB-231”
Mistake 9: From line 193 to 206 and 226 to 256 – double space between lines and no justification of the text
Coreection: Line Spacing chage to 13 pounds of the whole text
Mistake 10: 233 similar mechanism of action (Figure 2).
Coreection: 233 “(Figure 2)” was changed to “ (Figure 6)”
Mistake 11: 250 Fig. 2 Growth inhibitory- and cytotoxic activities of compounds 7d, 7k, and 7q across a panel of the JFCR39 cell
Coreection: 250 “Figure 2” was changed to “ Figure 6”
Additional revision
Mistake 12:
197 Compound 7d shows a notable cytotoxicity against the breast (BSY-1, LC50 = 0.82 µM), CNS (SF-539, LC50 = 0.60
Coreection: “7d” was changed to “ 7k “
Mistake 13:
200 Compound 7k shows a mentionable cytotoxicity for the CNS (SF-539, LC50 = 0.63 µM, SNB-75, LC50 = 0.93 µM),
Coreection:“7k” was changed to “7q”
Mistake 14:
235 = 0.7 for both). Similarly, compound 7k has a slight similarity to paclitaxel (r = 0.64). Compound 7l shows
Coreection: “7l” was change to “7q”.
Round 2
Reviewer 1 Report
While this work is fine in principle, it does not provide a significant
advance. To be of broad interest, the authors need to progress beyond
simple cytotoxicity assays. I have nothing further to add that can
assist with review of the current work.